# Implications of exclusive $J/\psi$ photoproduction in a tamed collinear factorisation approach to NLO

**Chris A. Flett[1]⋆, Alan D. Martin[2], Misha G. Ryskin[3] and Thomas Teubner[3]**

**1** Department of Physics, University of Jyväskylä, P.O. Box 35, 40014 University of Jyväskylä, Finland and Helsinki Institute of Physics, P.O. Box 64, 00014 University of Helsinki, Finland
**2** Institute for Particle Physics Phenomenology, Durham University, Durham, DH1 3LE, U.K.
**3** Department of Mathematical Sciences, University of Liverpool, Liverpool, L69 3BX, U.K.

⋆ cflett@jyu.fi

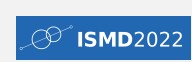
## Abstract

We discuss exclusive $J/\psi$ photoproduction, initially in conventional collinear factorisation at NLO and then subsequently in a refined approach with a programme of low $x$ resummation and implementation of a crucial low $Q_0$ subtraction included. We compare and contrast predictions in both frameworks and remark about the possibility to constrain and ultimately determine the low $x$ and low scale gluon PDF, emphasising the significance of this for future global PDF analyses.

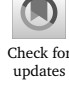
## 1 Introduction

The multitude of *inclusive* Deep Inelastic Scattering (DIS) experimental data from colliders past and present do not well constrain the global parton distribution function (PDF) analyses at small momentum fractions $x$ and factorisation scales $\mu^2$. Fig. 1 (left) shows, for example, the gluon PDFs obtained in the MSHT20 [1], NNPDF3.1 [2] and CT18A [3] NLO analyses as a function of the parton momentum fraction $x$ at a fixed low scale $\mu = 1.55$ GeV. At this scale, it is clear that as $x$ gets smaller, the uncertainties attributed to the gluon PDF become larger, and the extractions are consistent with an arguably unphysical decreasing gluon density as well as a vanishing one. This is primarily due to the lack of experimental data constraints in the global fits in this small $x$ and small $\mu$ domain.

As is evident in Fig. 1 (right), there also exists a sizeable uncertainty for larger $x > 0.1$ in the region where the valence quark PDFs dominate. This is not visible in the log-linear display of Fig. 1 (left) but becomes so if we take ratios of the sets normalised to e.g. the CT18A set, as shown in Fig. 1 (right). This large uncertainty has implications for quark PDF flavour separation and Beyond-The-Standard-Model searches but, in these proceedings, we will be

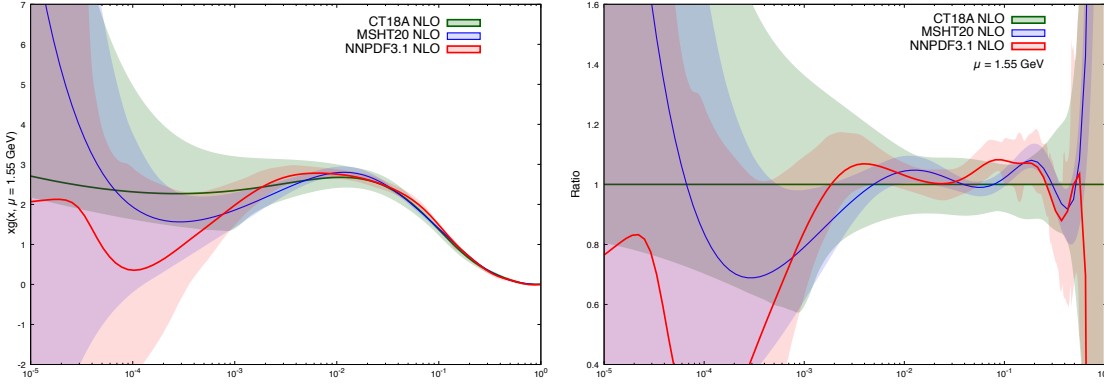

Figure 1: *Left:* Comparison of the absolute values of the NLO gluon PDFs obtained in various global analyses [1–3] at the scale $\mu = 1.55$ GeV. *Right:* Comparison of ratios of the NLO gluon PDFs normalised to the CT18A set, emphasising the large uncertainties present at low- and high-$x$.

concerned with only the small $x$ and small $\mu$ domain. In particular, we study *exclusive* $J/\psi$ photoproduction, $\gamma p \to J/\psi p$, and explain why the data for this process from HERA and LHC can be used in the global PDF analyses to determine the gluon PDF down to $x \sim 3 \times 10^{-6}$ and factorisation scales $\mu^2 = O(M_{J/\psi}^2/4)$, where $M_{J/\psi}$ is the mass of the $J/\psi$. As we will describe, this is complicated theoretically as, due to the off-forward, or skewed, parton kinematics, the process is sensitive to the generalised parton distribution function (GPD) rather than the usual forward one and, moreover, within the perturbative QCD (pQCD) approach to next-to-leading order (NLO), there is a large factorisation scale dependence exhibited by the $\gamma p \to J/\psi p$ hard matrix element. Nevertheless, we will show that this scale dependence can be largely alleviated by supplementing standard collinear factorisation with resummation of a class of large double logarithms [4] and implementation of a crucial low $Q_0$ subtraction [5], where $Q_0$ is the PDF input scale. We refer to this as a 'tamed' collinear factorisation approach in the following. Furthermore, the GPD can be related to the PDF via the so-called Shuvaev integral transform [6], which admits an accuracy of $O(x)$ at NLO in pQCD and so, on a practical level, becomes a reliable means to ascertain the behaviour of skewed partons at small $x$.

After describing our general framework in Section 2, we compare and contrast exclusive $J/\psi$ photoproduction to NLO in the conventional and tamed collinear factorisation approaches in Section 3. Then, in Section 4, we discuss the implications of this tamed approach for the $\overline{\text{MS}}$ gluon PDF and end, in Section 5, with concluding remarks.

## 2 Framework

The exclusive $J/\psi$ photoproduction process, $\gamma p \to J/\psi p$, proceeds via the fluctuation of a quasi-real photon into a $c\bar{c}$ pair, which hadronises into the final state $J/\psi$ meson through a two-parton exchange mechanism, see Fig. 2. In the Bjorken limit at leading twist, the net momentum fraction along the lightcone direction $P^+$ is $2\xi$, where $\xi$ is the skewedness parameter, and so the parton momentum fractions can be parametrised as $X + \xi$ and $X - \xi$. The $c\bar{c}$ to $J/\psi$ transition is described within LO Non-Relativistic QCD (NRQCD), with the relative quark velocity in the outgoing $c\bar{c}$ pair set to zero and with $M_{J/\psi} = 2m_c$, where $m_c$ is the mass of the charm quark. The first relativistic correction has been studied and shown to be small once the LO non-perturbative matrix element is normalised to the leptonic decay width of the $J/\psi$ [7].

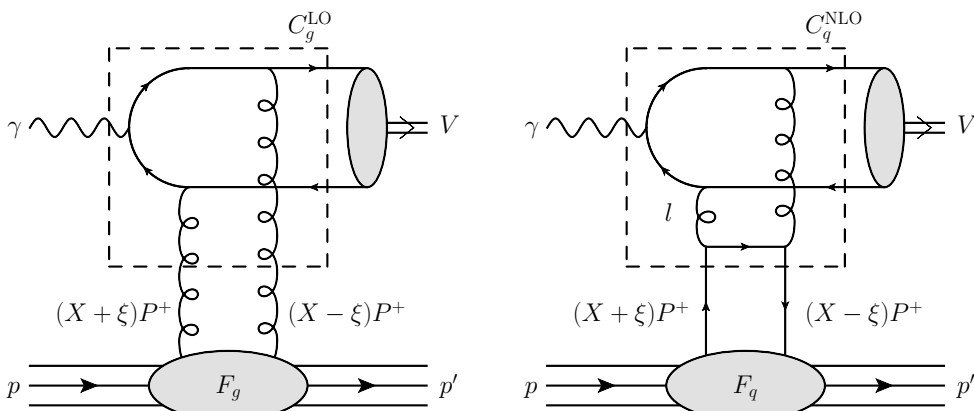

Figure 2: Examples of a LO contribution (*left*) and NLO quark contribution (*right*) to the exclusive process $\gamma p \rightarrow J/\psi p$. Here, the momentum $P \equiv (p + p')/2$ and $l$ is the loop momentum. The momentum fractions of the left and right input partons are $x = X + \xi$ and $x' = X - \xi$ respectively; for the gluons coupled directly to the on-shell heavy-quark pair, we have $x' \ll x$ and so $x \simeq 2\xi$.

The amplitude for exclusive $J/\psi$ photoproduction in the collinear factorisation framework at NLO can therefore be written as

$$A(\xi) \sim \sqrt{\langle O_1 \rangle_V} \int_{-1}^{1} \frac{\mathrm{d}X}{X} \left[ C_g(X, \xi, \mu_R, \mu_F) F_g(X, \xi, \mu_F) + C_q(X, \xi, \mu_R, \mu_F) F_q(X, \xi, \mu_F) \right], \quad (1)$$

where $\langle O_1 \rangle_V$ is the LO non-perturbative NRQCD matrix element, $C_{g,q}$ are the NLO gluon and quark coefficient functions [8] and $F_{g,q}$ are the NLO gluon and quark singlet GPDs. The renormalisation and factorisation scales are denoted by $\mu_R$ and $\mu_F$, respectively. At LO, there is only the gluonic term while at NLO there is a contribution from the quark sector, too. The GPDs at sufficiently small values of $x, \xi$ are reliably obtained via the Shuvaev transform [6], with the forward PDFs taken from the LHAPDF6 interface [9].

## 3 Exclusive $J/\psi$ photoproduction in collinear factorisation

In the conventional collinear factorisation approach to NLO in the $\overline{\mathrm{MS}}$ scheme, the exclusive $J/\psi$ photoproduction suffers from a large factorisation scale dependence and the NLO result is larger and of opposite sign to the LO one, suggesting a poor perturbative stability in $\alpha_s$. This is illustrated in Fig. 3, which shows $\mathrm{Im}\, A/W^2$ as a function of $W$, the $\gamma p$ centre-of-mass energy. We use state-of-the-art CT18ANLO input partons and show the LO and NLO scale variation bands for $\mu_F^2 = \mu_R^2 = Q_0^2, m_c^2, 2m_c^2$. It is clear that the factorisation scale dependence worsens at NLO and that in general this process exhibits a strong sensitivity with respect to scale variations. However, inspection of the amplitude in the high-energy limit, $W^2 \gg M_{J/\psi}^2$, shows that logarithmically enhanced contributions at small $x$ can spoil the perturbative convergence.

In particular, as the first ingredient of our tamed collinear factorisation approach, we consider the resummation of a class of large double logarithms of the form $\alpha_s \ln(1/\xi) \ln(\mu_F^2/m_c^2)$ [4]. In the high-energy limit, the NLO amplitude becomes directly proportional to a logarithm of this kind and so the judicious factorisation scale choice $\mu_F = m_c$, motivated by the structure of the amplitude in the high-energy asymptotics, eliminates this large NLO contribution, as well as simultaneously resuming this important class of double log-

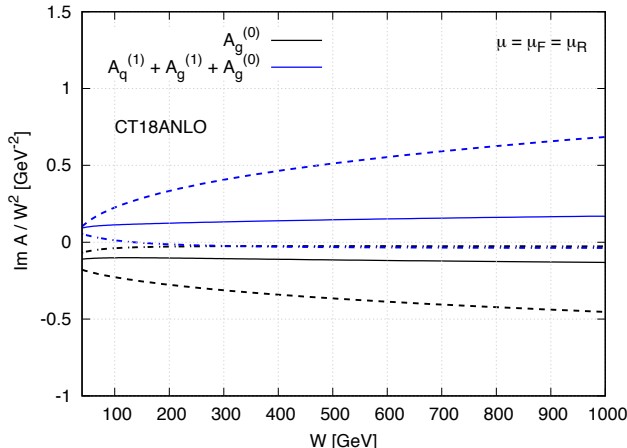

Figure 3: LO (black) and LO+NLO (blue) contributions to Im $A/W^2$ generated using CT18ANLO [3] global partons with $\mu_F^2 = \mu_R^2 = Q_0^2, m_c^2$ and $2m_c^2$ (from bottom to top). Im $A$ is the imaginary part of the amplitude.

arithmically enhanced contributions.[1] The setting of the factorisation scale translates into a shifting of terms between the higher-order contributions. For example, at some factorisation scale $\mu$, the quark initiated contribution is part of the NLO hard matrix element while at another factorisation scale $\mu'$, it is absorbed into the LO result. Our ideology is to find a so-called 'optimal' scale that removes the largest contribution from the NLO correction and that, as explained above, is given by the choice $\mu_F = m_c$. The effect of the scale change from a given $\mu_F$ to $\mu_f$ is driven by DGLAP evolution, in which generalised (skewed) splitting functions $V$ are present, and the change in the LO amplitude is given by

$$A^{(0)}(\mu_f) = \left( C^{(0)} + \frac{\alpha_s}{2\pi} \ln\left(\frac{\mu_f^2}{\mu_F^2}\right) C^{(0)} \otimes V \right) \otimes F(\mu_F), \qquad (2)$$

while to NLO it is

$$A(\mu_f) = C^{(0)} \otimes F(\mu_F) + C^{(1)}(\mu_F) \otimes F(\mu_f). \qquad (3)$$

Figure 4 (left) shows the effect of this double logarithmic resummation. Here, the factorisation scale $\mu_F = m_c$ is fixed numerically and variations are made with respect to the residual factorisation scale $\mu_f$, introduced above. As expected, the scale dependence improves noticeably but still not satisfactorily as, for example, the NLO result is still of the opposite sign to the LO one. The variations can be eased further by considering an additional class of contributions which take the shape of power corrections $O(Q_0^2/\mu_F^2)$ to the standard coefficient functions [5]. These constitute the other ingredient in the tamed approach and arise through the implementation of a crucial '$Q_0$' subtraction which cures an inherent double counting in the standard collinear factorisation prescription within the $\overline{\text{MS}}$ scheme. Explicitly, in this procedure, the contribution from scales $l^2 < Q_0^2$ circulating in the Feynman diagram virtual loops are subtracted from the full NLO coefficient function. This avoids the double counting in the convolution of the coefficient functions with the input GPDs at some $Q_0$. Figure 4 (right) shows the effect of the double logarithmic resummation together with the $Q_0$ subtraction. The scale variation has dramatically improved and the results are indicative of a perturbative convergence, suitable

---

[1]Note that the alternate $k_t$-factorisation scheme intrinsically resums the same class of large logarithms, but in this framework only a subset of NLO corrections belonging to the equivalence class of gluon-ladder diagrams are considered.

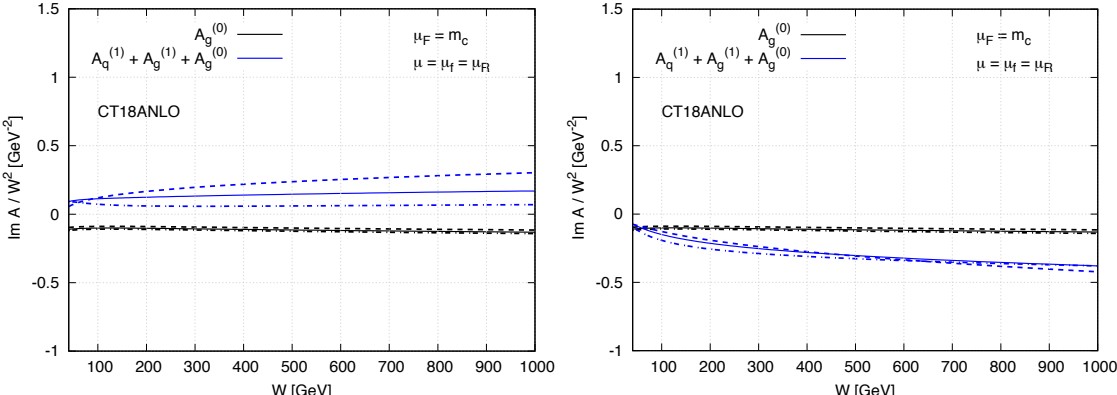

Figure 4: As in Fig. 3 but now with $\mu_F = m_c$ fixed before (*left*) and after (*right*) the double counting correction has been implemented, see text. Scale variations are now made with respect to $\mu_f$.

for a reliable comparison with experimental data. We now provide some further discussion of the $Q_0$ subtraction.

## 4 Sensitivity to the $\overline{\text{MS}}$ scheme gluon PDF

It is important to emphasise that we remain in the $\overline{\text{MS}}$ scheme while working with $Q_0$ subtracted coefficient functions to NLO accuracy. Specifically, the subtraction does not affect the infrared collinear or ultraviolet divergence renormalisation procedures and the soft singularity at $l = 0$ is removed by subtracting off the LO part of the NLO coefficient function before the integral over the loop momentum from 0 to $Q_0$ is performed. It is precisely this finite contribution from $l < Q_0$ that is subtracted to avoid the double counting inherent within the $\overline{\text{MS}}$ scheme. To NLO this subtraction procedure does not give rise to a new set of splitting functions and PDF evolution that otherwise a new scheme would entail. We remark that this subtraction is fundamentally ubiquitous and should be employed to correct for the double counting in all $\overline{\text{MS}}$ coefficient functions, see [10] for the procedure applied to inclusive DIS and Drell-Yan production. However, as these additional corrections are of the form $O(Q_0^2/\mu_F^2)$, when the typical factorisation scale of the process $\mu_F \gg Q_0$, the effects of this subtraction are not visible and only become relevant for lower scale processes, like exclusive $J/\psi$ photoproduction discussed here.[2]

## 5 Conclusion and final remarks

In these proceedings, we have presented a comparison of approaches of exclusive $J/\psi$ photoproduction to NLO within collinear factorisation and shown that a programme of resummation together with consideration of additional crucial power corrections solves the long known problem of the large scale sensitivity exhibited by the exclusive $J/\psi$ photoproduction process. In an earlier analysis [11], we showed that analogous predictions to those presented in Section 3 gave cross-section predictions that agreed favourably with the HERA data for exclusive $J/\psi$ photoproduction, and with good scale stability. The large difference between

---

[2]For exclusive $J/\psi$ photoproduction, $\mu_F = O(m_c)$. In our approach, as discussed in Section 2, $\mu_F = m_c$ and so the ratio $Q_0/m_c = \text{O}(1)$ and the subtraction is therefore crucial.

the predictions obtained using a variety of different global PDFs in the larger $W$ LHC regime, together with their reconciliation in the lower $W$ HERA regime, motivated the extraction of a low $x \sim 3 \times 10^{-6}$ and low $\mu \sim m_c$ gluon PDF via two approaches which were shown to be consistent [12]. We emphasised that, as the global fit PDF errors are so much greater than the exclusive $J/\psi$ experimental data uncertainties and the underlying scale uncertainty associated with the theory prediction in our tamed collinear factorisation framework, the exclusive $J/\psi$ data are in a position to readily improve the global PDF analyses. In this way, the exclusive $J/\psi$ data at large $W$ from LHC will constrain the low $x$ and low $\mu$ behaviour of the gluon PDF, while the data at lower $W$ from HERA can provide constraints in the regime where there is the aforementioned multitude of data constraints from inclusive, as well as diffractive, DIS experiments spanning a wide range of momentum fractions $x$ and scales $\mu$.

## Acknowledgements

**Funding information** CAF is supported by the Helsinki Institute of Physics core funding project QCD-THEORY. The work of TT is supported by the STFC Consolidated Grant ST/T000988/1.

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
