# Peer review of "Implications of exclusive $J/ψ$ photoproduction in a tamed collinear factorisation approach to NLO"

_SciPost Physics Proceedings, doi:SciPost Phys. Proc. 15, 005 (2024)_

## Round 1 · Referee Report · Anonymous (Referee 1) · 2023-1-6

Report

The author has written a clear and concise summary of their results which they presented at ISMD 2022. It therefore meets the criteria for publication.

Specifically the author has described results which demonstrate that including small-x resummation improves the stability of predictions for exclusive $J/\psi$ photo production. The implementation of low $Q_0$ subtraction leads to a further improvement in stability. The report motivates the study, setting it in the context of known results and demonstrating the potential improvement of their method.

Requested changes

  1. Is a sum over quark flavour missing in the second term of equation (1)?

---

## Editorial Decision

published